# Recently Discovered Thick Bentonite Bed Hosted by the Lithothamnium Limestones (Badenian) in the Polish Part of the Carpathian Foredeep: The Evidence for Volcanic Origin

Katarzyna Górniak [1,*], Tadeusz Szydłak [1], Piotr Wyszomirski [2,3], Adam Gaweł [1] and Małgorzata Niemiec [1]

[1] Faculty of Geology, Geophysics and Environmental Protection, AGH University of Science and Technology, Al. Mickiewicza 30, 30-059 Kraków, Poland; szydlak@agh.edu.pl (T.S.); agawel@agh.edu.pl (A.G.); mniemiec@student.agh.edu.pl (M.N.)

[2] Faculty of Materials Science and Ceramics, AGH University of Science and Technology (Emeritus), Al. Mickiewicza 30, 30-059 Kraków, Poland; pwysz@agh.edu.pl

[3] State Higher Vocational School in Tarnów, Ul. Mickiewicza 8, 33-100 Tarnów, Poland

\* Correspondence: gorniak@agh.edu.pl

**Abstract:** In this paper, we discuss the hypothesis on the volcanic origin of the precursor sediments for a thick (0.6 m) clay bed, hosted by the sequence of lithothamnium limestones of the Pińczów Formation. Combined X-ray powder diffraction, imaging methods (optical and electron microscopy), and chemical analysis were used to document the volcanic markers, which were preserved in the rock studied. The results obtained show that the clay bed discussed is bentonite in origin. This bentonite, which can be called Drugnia Rządowa bentonite, is composed almost entirely of montmorillonite with little admixtures of quartz and biotite. A small amount of calcite is present, but only in the top of the bed. Despite that, the bentonite contains nothing but clay material—it is a model example of entirely altered pyroclastic rock, which retains texture originally developed in volcanic glass fragments and reveals the preserved original features of the precursor fallout pyroclastic deposits (rhyolitic in character). The thick bentonite beds, discovered for the first time within the Badenian lithothamnium limestones of the Pińczów Formation, can be considered as a record of a violent, explosive volcanic event related to the closure of the Outer Carpathian basin and the development of the Carpathian Foredeep.

**Keywords:** bentonite; imaging; Badenian; petrology of bentonites; volcanic markers; northern Carpathian Foredeep; lithothamnium limestones; Drugnia Rządowa; Poland



## 1. Introduction

The presence of a thick, carbonate-free clay bed (several dozen centimeters in thickness) in a marine limestone sequence raises questions about the origin of such lithological diversity. It is obvious that the conditions of sedimentation of the limestone and clays are different (see [1,2]). The clays that occur in limestone successions are often regarded as a result of the alteration of accumulated volcanic glass to clay minerals in a marine environment (e.g., [3]). Consequently, the clay beds composed of volcanogenic clays are thought to represent a record of a synsedimentary volcanic activity delivering the pyroclasts to the sedimentary basin of the limestone. However, the hypothesis that the clay minerals are the product of the alteration of volcanic material and the resulting clay interbeds in limestone successions are bentonites in origin needs to be proved. The task is easy when shards of volcanic glass are retained in the clay beds, but a challenge when they are entirely altered [4]. Fortunately, there are some petrographic features of clays, which allow us to relate the origin of clay beds with the alteration of volcanogenic material. These features are: (1) volcanic glass shard textures are preserved despite the glass shards being entirely replaced by clay (e.g., [5]); (2) textures of volcanic glass shards have not been preserved by the clay, but the nature of the crystal pyroclasts (morphology and composition) gives clear

evidence of their origin (e.g., [6,7]); and (3) the nature of clay is peculiar, i.e., the mineralogy is usually dominated by one clay type (e.g., smectite), which reveals the morphology of an authigenic mineral and has specific crystal–chemical features, e.g., the layer charge in smectite is usually located in its octahedral sheets (e.g., [3]).

Bentonites are defined by petrologists as clays formed in seawater due to the alteration of vitric components of pyroclastic materials to smectite (e.g., [3]). The bentonites that originate from the alteration of a submarine fallout ash usually form thin layers, which are a millimeter to a few centimeters in thickness. Thick beds, about one meter thick, related to large volcanic eruptions, are rarely preserved in the geological record. The significance of bentonites in geology and industry is widely elaborated in literature. Bentonite beds are used, e.g., as timelines and consequently, a tectonic interpretation of sedimentary patterns is possible (see review in [8]). Such thick layers are also important raw materials for many branches of industry (e.g., [9–11]).

The study area is located in the northern part of the Carpathian Foredeep, at the edge of the Western European Platform (Figure 1). The development of the Carpathian Foredeep is closely related to tectonic activity accompanied by volcanic eruptions. The latter have been recorded as numerous volcanic fields located in the inner part of the Carpathian arc (e.g., [12–14]). The volcanic eruptions accompanying sedimentation in the Carpathian Foredeep have been recorded as pyroclastic interbeds, which occur in the whole sequence of the Miocene deposits in the Carpathian-Pannonian region. Although the timing of these pyroclastic deposits and their correlation with source volcanoes were largely elaborated in the literature, there are still many disputable problems (e.g., [15]; Polish part of the Carpathian Foredeep [16]; review [17]).

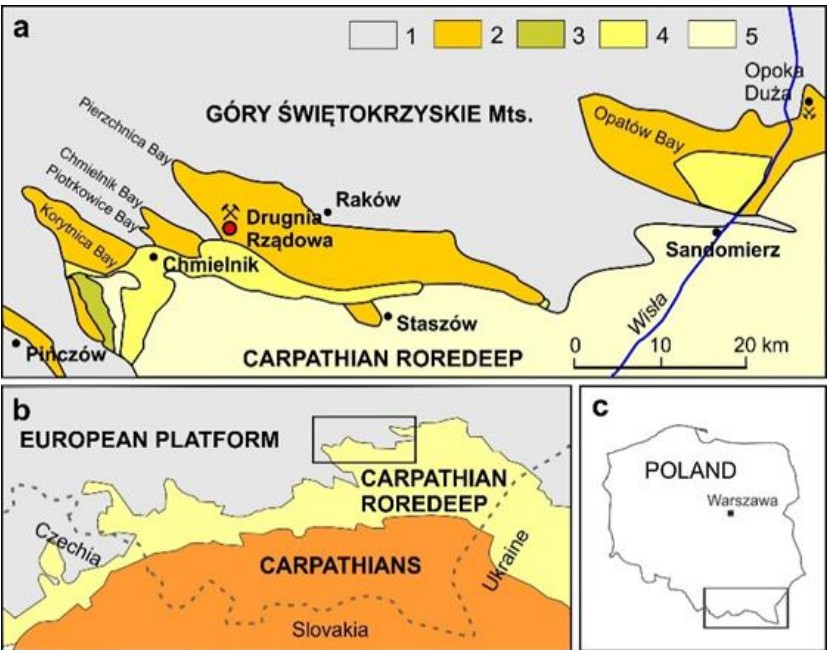

**Figure 1.** Location of the outcrop studied (limestone open pit, Drugnia Rządowa) in the northern margin of the Carpathian Foredeep area (**a**). The sketch is based on the geological map drawn to a scale of 1:200,000 published by the Polish Geological Institute [18] and the geological data from Radwański [19], simplified. Explanations: 1—rocks older than Miocene; 2–5—Miocene: 2—Pińczów Formation, 3—Krzyżanowice Formation, 4—Chmielnik Formation, 5—Machów Formation. The situation of the study area in the Carpathian Foredeep (**b**) and the location of the Carpathian Foredeep in Poland (**c**) are show at the bottom of this figure.

Currently, in the northern, marginal part of the Carpathian Foredeep, Sarmatian bentonites and montmorillonite clays are the best known. They were discovered and excavated near Chmielnik until the 1990s. The montmorillonite clays were mostly in use

as bentonite raw materials because they were of a few meters in thickness, whilst two accompanying bentonite interbeds were only of a few to a dozen centimeters thick [20]. Industrial bentonite deposits in Poland are rare; therefore, since the closure of the mining activity in Chmielnik, Polish companies use bentonite raw material, which comes mostly from Slovakia and is excavated in the Central Slovakia Volcanic Field located in the inner part of the Carpathian arc [21,22].

This paper focuses on a thick (0.6 m) clay bed occurring in the marginal, northern part of the Carpathian Foredeep within the Badenian deposits of the Miocene nearshore facies represented by the lithothamnium limestones. The clay bed was recently discovered in an open pit in which lithothamnium limestones are excavated. This discovery raised the question of how to explain the origin of the bed. If the clay bed was bentonite in origin, due to its thickness, the layer would be worthy of detailed study. Only one thick bentonite bed was earlier described in the Polish part of the Carpathian Foredeep. The Opoka Duża bentonite bed up to 90 cm (average 40 cm) in thickness was mined nearby Annopol in the 1930s. Czarnocki [23] reported the occurrence of this bed within a sequence of marine sandstones underlying lithothamnium limestones. It is worth mentioning that the Opoka Duża mine was one of Poland's first domestic exploited economic bentonite deposits. In the 1950s and 1960s, a few thin bentonite interbeds from the Sarmatian deposits were also excavated in the northern part of the Carpathian Foredeep. The location of the open-pit mines and the descriptions of the deposits were published by Fijałkowska and Fijałkowski [24]. In this region there were no previous reports of bentonite hosted by lithothamnium limestones. However, clay aggregates, presumably originated from the alteration of volcanic glass shards, dispersed among red algae remnants in the lithothamnium limestones were noticed [25].

The aim of this paper is to verify that the clay bed studied has a volcanic origin. In order to decipher traces of volcanic activity recorded in the clay bed, combined petrographic methods, X-ray diffraction and imaging (optical and electron microscopy), were applied. Petrographic studies, especially imaging in detail, are scarce in the literature related to bentonites from the Carpathian Foredeep. This paper links two problems. The results obtained may give some suggestions concerning the proper use of these clays being potentially peculiar raw material and could provide some arguments to the discussion about the geological history of the Carpathian Foredeep. The presence of a thick bentonite bed within lithologically variable Miocene deposits could be an important finding for better understanding the tectonics and sedimentation conditions in the Carpathian Foredeep basin. This work could bridge the gap in research of the volcanic activity accompanying the development of the Carpathian Foredeep.

## 2. Geology

The outcrop studied is located in southern Poland, at the northern margin of the Carpathian Foredeep (Figure 1). The Carpathian Foredeep is a large geological structure, which extends along the Carpathian arc. In the Polish part, the foredeep is about 300 km in length and about 100 km in width. The thickness of the molasse deposits accumulated in the basin reaches about 3000 m in its central part, and does not exceed several hundred meters at the northern margin (*fide* [26]). The Carpathian thrust front marks the southern edge of the Carpathian Foredeep, and its northern edge is erosive [27]. The Carpathian Foredeep was created due to movement towards the north and over-thrusting of the Carpathian nappes onto the foreland plate. In response to the loading, the foreland flexed downward in front of the mountain belt and a foreland basin was formed (e.g., [27,28]). The northern, marginal part of the Carpathian Foredeep was intensively studied for decades. A review of the papers published on geology of this region can be found in [29].

The clay bed studied occurs within the deposits of a transgressive succession, which is Badenian in age. The Badenian transgression arrived at the northern margin of the basin in a varied terrain relief (southern and south-eastern slopes of the Holy Cross (Świętokrzyskie) Mountains). Consequently, a seashore similar to the Dalmatian seashore in character was

formed [19,30]. The study area is located in the western part of the southern slopes of the Holy Cross Mountains. In this region, Radwański [30] recognized the zone of four large rocky bays (i.e., Pierzchnica, Chmielnik, Piotrowice and Korytnica bays), and the forebay zone (i.e., the nearshore part of the basin south of the bays) (Figures 1 and 2), which differ in sedimentary conditions and resulting sediments. The Miocene lithostratigraphy in these zones is presented in Figure 2. The basement of the Miocene deposits in the region is formed by the Paleozoic and Mesozoic rocks of the European Platform (e.g., [27,28,30]). The Miocene sequence is represented by the Badenian deposits, which belong to the Trzydnik Formation, Pińczów Formation and Krzyżanowice Formation, and the Badenian-Sarmatian deposits, called the Machów Formation and the Chmielnik Formation [31].

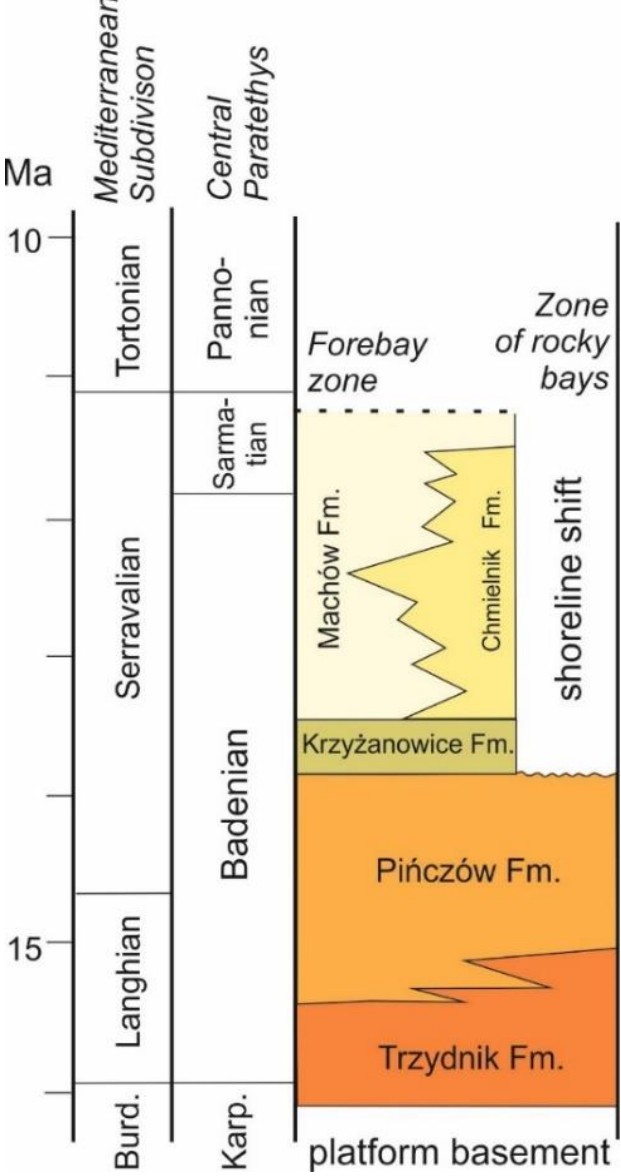

**Figure 2.** Stratigraphic column of the Miocene in the northern margin of the Carpathian Foredeep area based on Alexandrowicz et al. [31] and Czapowski [26], modified. The shoreline shift is after Radwański [30]. Expansion of the abbreviation: Fm.—Formation.

The Trzydnik Formation (Fm.), which was established at the very bottom of the Miocene succession, consists of sandstones and mudstones with lignite seams, up to a few dozen meters thick. In the zone of rocky bays, these deposits have been recognized only in the Korytnica Bay [32,33]. Bentonite beds have been spotted in this formation;

however, their thicknesses never exceed a few centimeters [24,25]. The Trzydnik Fm. is covered by the deposits of the Pińczów Fm. The Pińczów Fm. includes carbonates and siliciclastic rocks showing remarkable facies variability. Their thickness was estimated as a few dozen meters. The deposits of the Pińczów Fm. were discovered both in the zone of rocky bays and in the forebay zone [19,30,34–36]. Only in the forebay zone, at the top of the Pińczów Fm., a sequence of gypsum and anhydrites called the Krzyżanowice Fm. was recognized. These deposits reach up to 20 m in thickness. The Miocene sedimentation in the marginal, northern part of the Carpathian Foredeep was completed with the Badenian-Sarmatian deposits called the Chmielnik Fm. The Chmielnik Fm. consists of nearshore conglomerates, sandstones and mudstones, which include detrital limestone interbeds. The thickness of the formation yields about 30 m [26,37,38]. A deep-water equivalent of the Chmielnik Fm. is the Machów Fm. The Machów Fm. comprises calcareous mudstones with thin sandstone interbeds, which reach about 2500 m in thickness in the central part of the Carpathian Foredeep basin. The Chmielnik Fm. and the Machów Fm. laterally interfinger. Thin bentonite seams were discovered both in the Chmielnik Fm. and Machów Fm. [20,24,39,40]. It is worth mentioning that in Radwański's [30] opinion, the deposits of the Krzyżanowice Fm. and the Machów Fm. were never present in the rocky bays zone, because of the falling sea level during the Late Badenian period and the regression of the shoreline (Figures 1 and 2).

The Pińczów Fm., which hosts the clay bed studied here, has the largest spatial distribution among all marginal facies in the Carpathian Foredeep basin [34]. This formation is divided into several members, among which the most characteristic is the Lithothamnium Limestone Member of up to 25 m in thickness [34]. The lithothamnium limestones, commonly known as the Pińczów limestones, are a famous raw material used for centuries, especially for construction and sculptural purposes. Early scientific papers on the lithothamnium limestones were published at the end of the 19th century (*fide* [34]). The facies variability of the Pińczów limestones and the red algae from these limestones were the subjects of the studies performed by Studencki [34–36].

## 3. Material

The clay bed studied was discovered due to mining works in the lithothamnium limestone open pit located at Drugnia Rządowa, which has been operated by the Kopaliny Mineralne Sp. Z o.o. company (Pierzchnica, Poland) since 2018. The open pit is situated in the Pierzchnica Bay (Figure 1). The spatial distribution of the lithothamnium limestone facies in the Pierzchnica Bay was proposed by Czarnocki [41]. In Radwański's [30] opinion, the spatial distribution of the facies in this bay, as presented in the geological map of the above-mentioned author, was conceptual because the Miocene is very weakly exposed in this area. Therefore, the lithothamnium limestone open pit at Drugnia Rządowa, which was opened in the 1980s, is an important outcrop documenting the deposition of the Pińczów Fm. in the Pierzchnica Bay. The open pit works a flat-lying, weakly lithified variety of lithothamnium limestones and is dug on two benches. The thickness of the limestone succession, which is subject to extraction, is about 16 m. The clay bed studied was discovered at the bottom of the open pit. This bed follows the bedding plane of the hosting rocks and reaches 60 cm in thickness. The limestones, which underlay the clay bed, are better lithified than the overlying limestones. The exposure shown in Figure 3 is the subject of the current study. Field work and sampling were performed in October 2019. In this paper, preliminary results of the studies for the samples representing the bottom, middle and top part of the clay bed are presented (Figure 3).

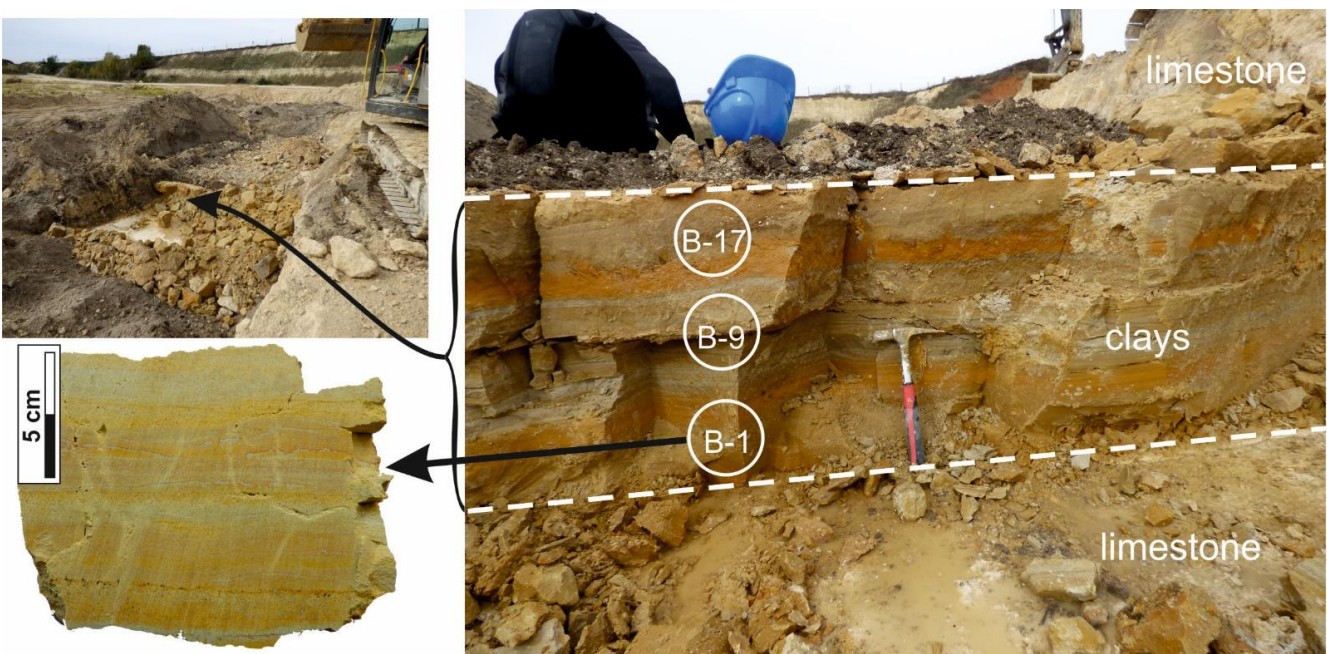

**Figure 3.** Clay bed hosted by the lithothamnium limestones in the Drugnia Rządowa open pit. B-1, B-9 and B-17 mark the sample positions throughout the bed thickness. The length of the hammer is 30 cm. A detailed description of the clay bed lithology is provided in the Results section.

## 4. Methods

The samples taken from the clay bed, as indicated in Figure 3, after naked eye examination in detail, were studied using X-ray powder diffraction (XRPD) and microscopic methods: optical and field emission scanning electron microscopy (FESEM). Integrated results of these studies allowed us to present the preliminary results on changes in mineralogy from the bottom to the top of the clay bed. The occurrence and the alteration degree of components of pyroclastic origin were possible to discover. Some remarks on the use of the clay studied as raw material can also be formulated.

The examination of the clay bed attributes on a centimeter to millimeter scale in the outcrop and in the hand specimens included recording the color, lithification rate, grain size, lamination style and description of non-clay components. The procedure proposed by Lazar et al. [42] was followed in order to standardize the field observations. Visual comparisons to the Munsell Rock-Color Chart [43] were applied in order to describe the color changes along the whole interval from the bottom to the top of the bed.

The XRPD studies were performed using a Rigaku SmartLab diffractometer (Rigaku Corporation, Tokyo, Japan) under the following conditions: graphite-monochromatized, CuK$\alpha$ radiation, operating voltage = 45 kV, current = 200 mA, step size = 0.05° 2θ and counting time = 1 s/step. The powder XRPD patterns were performed for qualitative phase analysis of the bulk samples. Random samples were prepared using the side-loading technique. Semi-quantitative estimations of mineralogy were conducted using the external standard method following the procedure described in Schultz [44]. The XRPD patterns were also recorded for oriented samples in order to identify clay minerals. The oriented samples were prepared by spreading suspensions of <2 μm fractions on glass slides (4 mg/cm$^2$) and drying them in an ambient atmosphere. The sedimentation process was employed in order to isolate <2 μm fractions. The XRPD patterns were obtained for air-dried oriented samples and then solvated with ethylene glycol at 60 °C for at least 16 h.

The optical observations were made using an Olympus BX 51 polarizing microscope (Olympus, Tokyo, Japan) on standard thin sections, which were 0.02 mm thick. Before mounting, the samples were impregnated with epoxy resin. The objective was to recognize

millimeter-scale lamination as well as to identify silt- and sand-size non-clay minerals, and to determine their abundances and distribution.

The FESEM studies were performed using an FEI Quanta 200 FEG (field emission gun) SEM (Field Electron and Ion Company, Hillsboro, Oregon, USA) equipped with secondary electron (SE) and backscattered electron (BSE) detectors and an energy dispersive spectrometer (EDS). Polished thin sections of 0.05 mm in thickness impregnated with epoxy resin and fresh rock fractures were analyzed. The BSE imaging of uncoated polished thin sections and the SE imaging of uncoated rock chips was performed at 20 kV in the low-vacuum mode. The objective was to reveal the presence and the alteration of volcanic glass shards to smectite. The morphology of smectite aggregates and the shapes and the alterations of non-clay minerals were examined. Chemical analyses were carried out with the EDS (point analyses) using a 50 s live time and 20 kV acceleration current potential.

## 5. Results

### 5.1. Hand Specimen Examination

The clays studied are, in general, grayish yellow (5Y 8/4 Grayish Yellow) and locally, in the middle portion of the bed, they are pale orange (10YR 8/6 Pale Yellowish Orange) and whitish (N9 White) in color (Figure 3). The wet rock, immediately after sampling, tended to behave in a ductile manner, while dried samples became brittle. Its dominant grains are clay-size, and silt- and fine sand-size materials are present, but in small amounts. The clays are subtly planar-parallel laminated. Sharply defined alteration is visible between colorful duller and lighter laminae (Figure 3). Some of them are also defined by grains coarser than clay-sized. Among these coarser grains are the easily recognizable biotite flakes up to 0.4 mm in size. The distribution of the biotite flakes varies throughout the bed thickness. Laminae seems to be obscured in the upper portion of the bed and sparse rhodoliths up to a few millimeters in size are observed. The yellow color of the clays indicates the presence of some iron oxides/oxyhydroxides in their composition.

### 5.2. X-ray Powder Diffraction

The XRPD traces of the bulk samples taken from the clay bed in the Drugnia Rzą-dowa open pit revealed that their mineral composition is dominated by smectite mineral (Figure 4). The traces contain intense and well defined basal $d_{001}$ peak of the smectite mineral at 15.3 Å. Additionally, in all XRPD traces, a weak but well-defined peak at 10.0 Å, which is attributed to micas, can be easily observed (Figure 4). Because this peak is not intense and the naked eye examination revealed localized biotite flakes, the biotite may be considered as the only mica occurring in the rocks studied. Besides the peaks attributed to smectite and biotite, weak reflections of quartz are visible in all XRPD traces. In addition to the above-mentioned minerals, calcite was found, but only in the sample taken at the top of the bed (sample B-17) (Figure 4).

Biotite, quartz and calcite are not frequent components of the rocks studied. Their total content is a low percentage and slightly varies throughout the thickness of the bed. It was roughly estimated that the biotite comprises 2% and quartz 1% in the samples taken at the bottom (sample B-1) and at the top (sample B-17) of the bed. About 7% of calcite was calculated in the sample taken from the top part of the bed (sample B-17), which occurs together with the above-mentioned admixtures of biotite and of quartz (2% and 1%, respectively). The middle portion of the bed (sample B-9) is composed almost exclusively of the smectite mineral accompanied only by about 1% of biotite, and trace amounts of quartz (Figure 4).

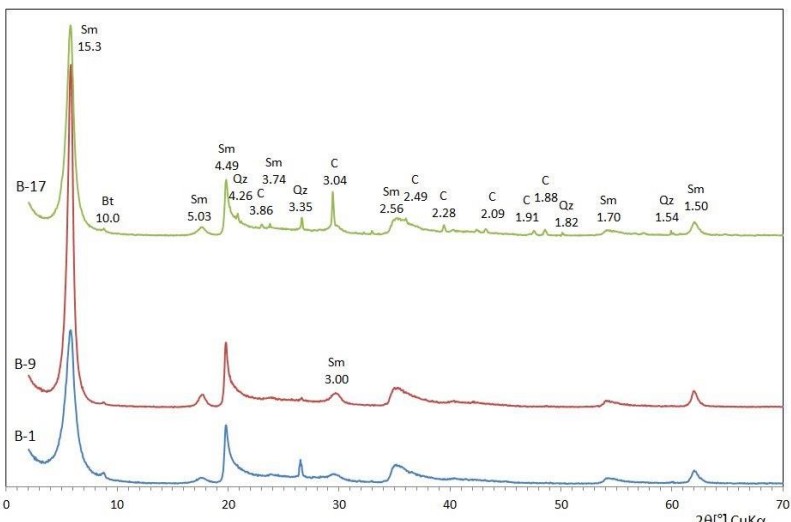

**Figure 4.** XRPD patterns of bulk samples of the Drugnia Rządowa clay bed. Samples: B-1—bottom, B-9—middle part, B-17—top of the bed. Key to the symbols: Bt—biotite, C—calcite, Q—quartz, Sm—smectite; $d_{hkl}$ in Å.

The XRPD traces of oriented samples (<2 μm fraction) recorded for the air-dried then solvated with ethylene glycol states showed that the basal 001 peak of the smectite mineral at 15.3 Å was shifted to 17.0 Å (Figure 5). This peak is strong and narrow. Its FWHM (Full Width at Half Maximum) value is 0.58° 2θ. The $d_{003}$ spacing at 5.60 Å indicates that the smectite mineral contains only swelling layers (see [45]). Consequently, the mineral studied can be called smectite. The XRPD pattern recorded for the sample in an ethylene glycol solvated state contains the set of the smectite basal reflections with d values equal to $d_{002}$ = 8.45 Å, $d_{003}$ = 5.60 Å, $d_{004}$ = 4.22 Å, $d_{005}$ = 3.35 Å and $d_{006}$ = 2.80 Å (Figure 5). These values deviate only slightly from a rational series.

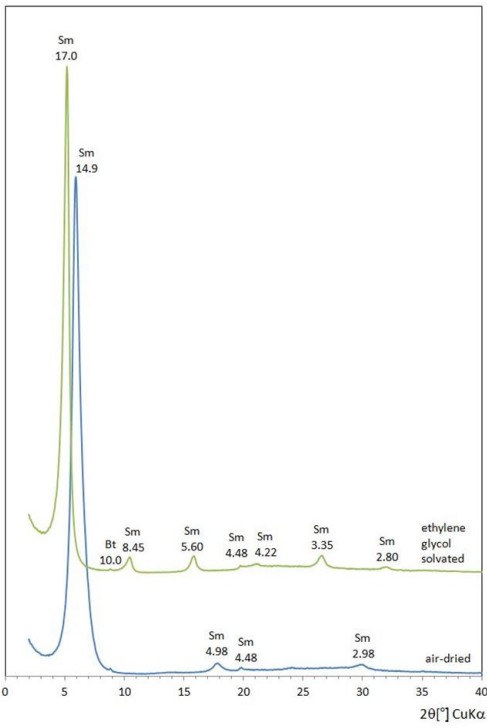

**Figure 5.** XRPD patterns of oriented, air-dried and ethylene glycol solvated, sample B-9 (<2 μm fraction). Key to the symbols: Bt—biotite, Sm—smectite; dhkl in Å.

### 5.3. Optical Microscopy

Under the optical microscope, the examined clays display partially preserved features of the original pyroclastic material. The groundmass with dispersed crystal pyroclasts can be distinguished (Figures 6–8). Localized laminae defined by iron oxides/oxyhydroxides with coarse crystal pyroclasts (Figures 6b and 7c) and without (Figure 7a) are visible. The former laminae are less than 1 mm, and the latter are usually about 0.2 mm thick.

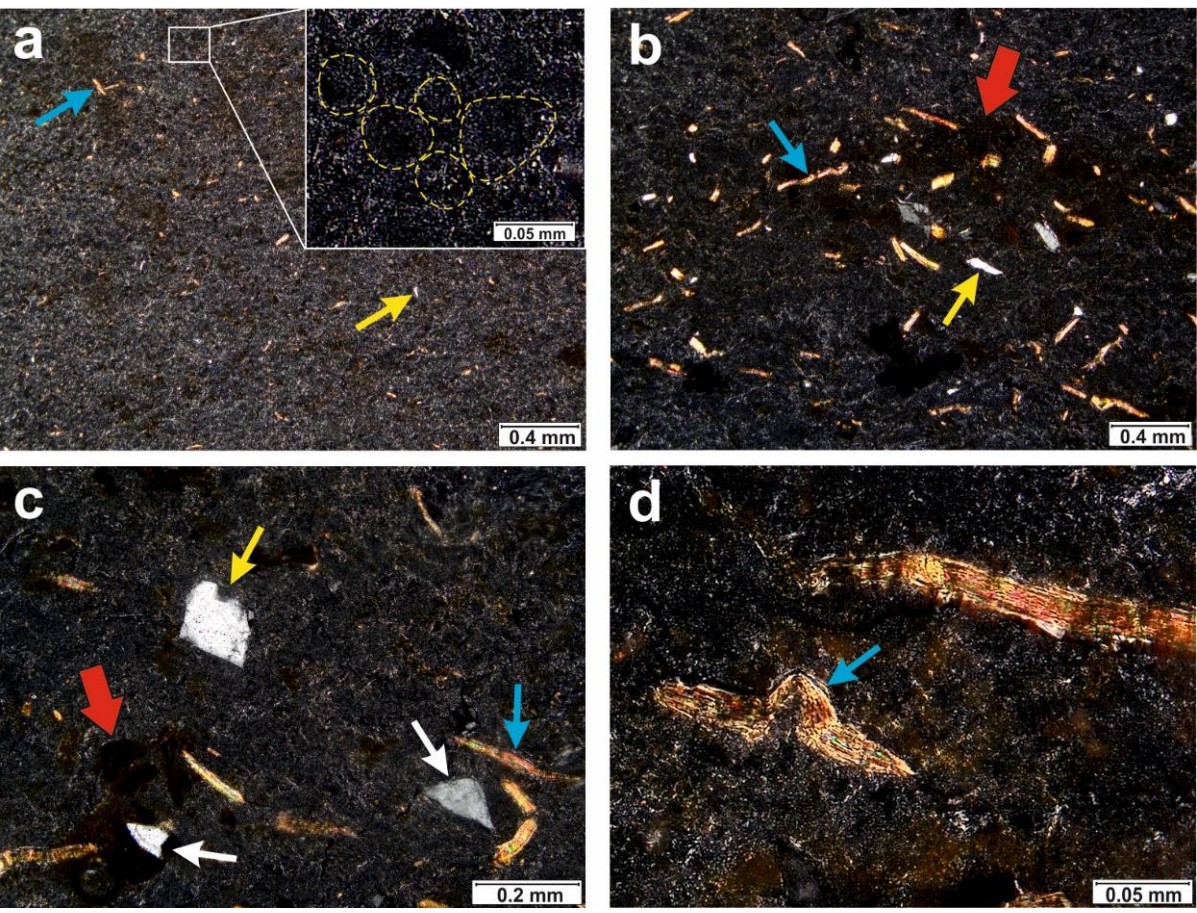

**Figure 6.** Thin-section micrographs (crossed polars) of the Drugnia Rządowa bentonite (sample B-1): (**a**) fine pyroclasts of biotite (blue arrow) and quartz (yellow arrow) scattered in groundmass composed of smectite pseudomorphs after volcanic glass shards. Note: enlargement of these pseudomorphs (outlined) in the area within the white box, (**b**) lamina defined by coarse pyroclasts of biotite (blue arrow) and quartz (yellow arrow). Note: iron oxide/oxyhydroxide brown in color accumulated in the vicinity of biotite flakes (thick red arrow), (**c**) embayed bipyramidal volcanic quartz crystal (yellow arrow), wedge-shaped quartz crystal chips (white arrows) and biotite flakes (blue arrow) revealed in a lamina defined by coarse crystal pyroclasts. Note: iron oxide/oxyhydroxide brown in color accumulated in the vicinity of biotite flakes (thick red arrow), (**d**) thin biotite flakes showing the diameter-to-thickness ratio about 1:10 accumulated in a lamina defined by coarse crystal pyroclasts. Note: ductile deformation of the flakes (blue arrow).

The groundmass is composed of clay with "ghosts" of volcanic glass shards, which are easily recognizable due to the different optical orientation of clay flakes in various pseudomorphs (Figure 6a). The size of these "ghosts" ranges from 0.05 to 0.1 mm. The groundmass is abundant and spotty because of localized concentrations of iron oxides/oxyhydroxides (Figures 6b and 7b).

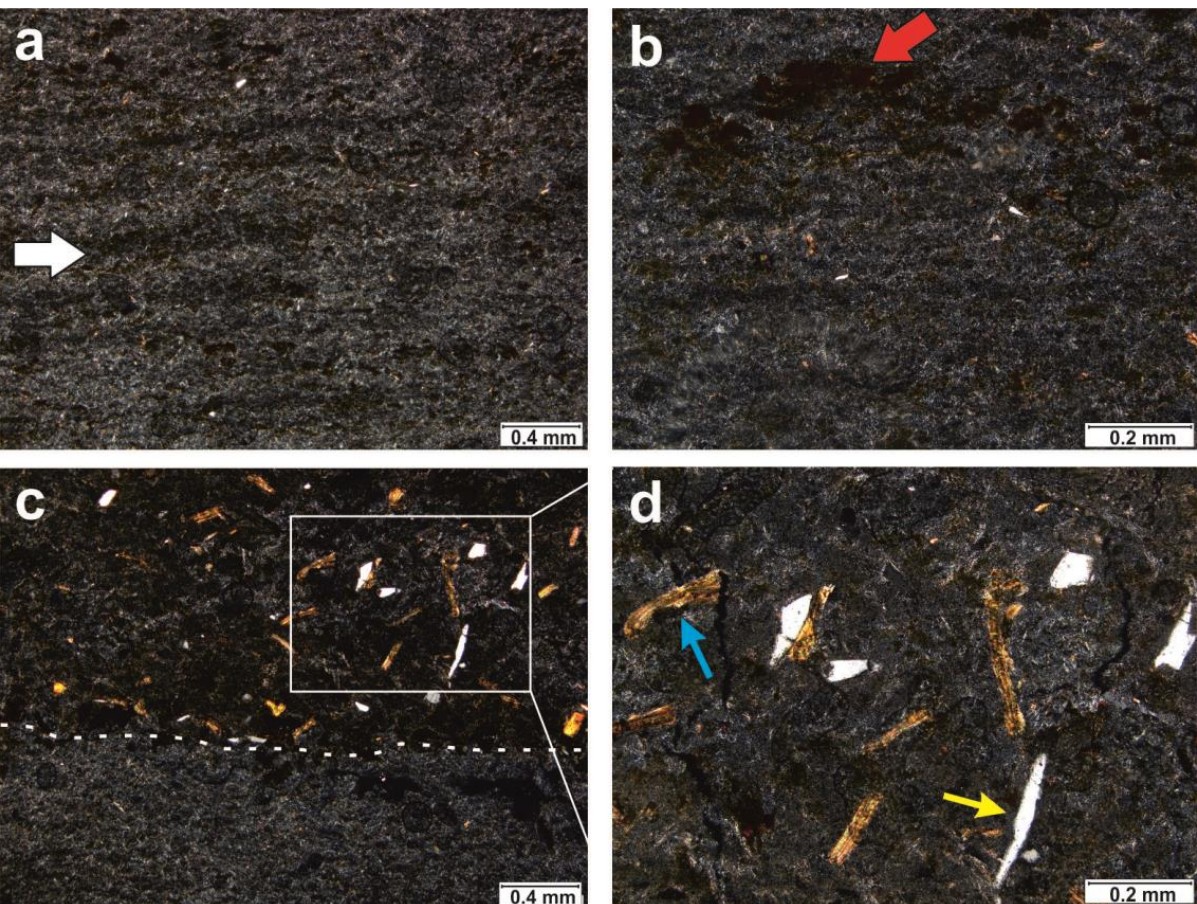

**Figure 7.** Thin-section micrographs (crossed polars) of the Drugnia Rządowa bentonite (sample B-9): (**a**) laminae defined by iron oxide/oxyhydroxide (thick white arrow). Note: laminae showing more birefringence in the section cut perpendicular to the bedding than in those cut parallel to it, which are related to the different orientation of smectite pseudomorphs after volcanic glass shards, (**b**) spotty concentrations of iron oxide/oxyhydroxide in the groundmass (thick red arrow), (**c**) lamina defined by coarse crystal pyroclasts of biotite and quartz. Note: sharp bottom of the lamina (dotted line), (**d**) enlargement of the area within the white box in (**c**) showing quartz crystal chips (yellow arrow) and thin biotite flakes (blue arrow). Note: wedge-shaped quartz crystal fragments in the middle part and splinter-shaped fragments in the bottom of the micrograph.

The crystal pyroclast assemblage mostly includes biotite, with small amounts of quartz. Both the biotite and the quartz vary greatly in size and distribution. The smaller grains (up to about 0.05 mm in diameter) are usually evenly disseminated throughout the thickness of the bed (Figure 6a), whilst the larger ones, ranging from 0.1 to 0.4 mm in diameter, occur mostly concentrated in laminae (Figures 6b–d and 7c,d). In the laminae defined by large biotite flakes and quartz grains, concentrations of iron oxides/oxyhydroxides were often observed. The distribution of these laminae over the entire thickness of the bed is uneven.

The biotite, both from the groundmass and from the laminae, usually forms thin flakes with a thickness to diameter ratio of about 1:10. The average flake diameter is 0.2 mm. The tiniest individuals are several micrometers in size, whilst the largest ones reach up to 0.4 mm in dimension. The majority of the flakes show ductile deformation, which are often very advanced (Figure 6d). Usually, biotite is fresh with a brown color when seen in plane polarized light. In the vicinity of some large biotite flakes, spotty concentrations of iron oxides/oxyhydroxides, light brown in color, are visible (Figure 6b).

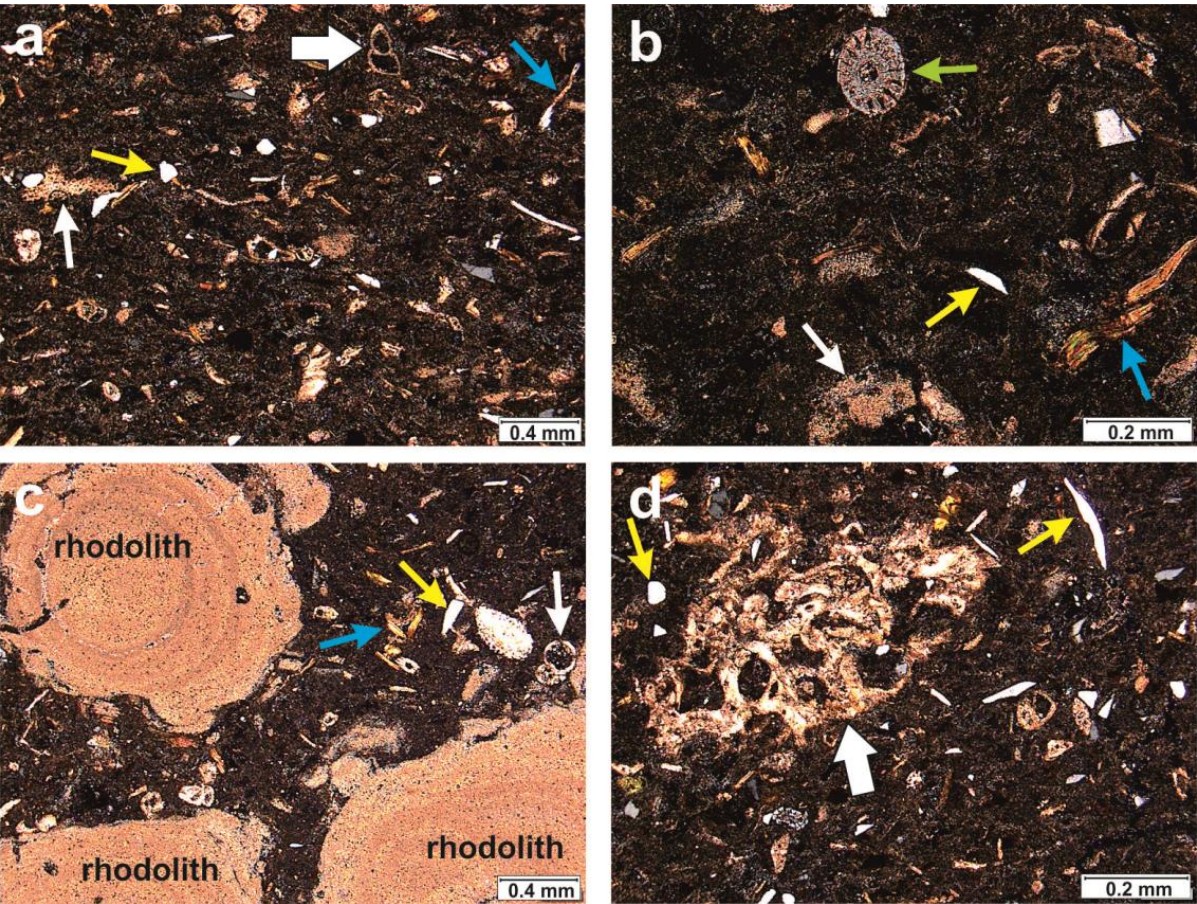

**Figure 8.** Thin-section micrographs (crossed polars) of the Drugnia Rządowa bentonite (sample B-17): (**a**) crystal pyroclasts of biotite (blue arrow) and quartz (yellow arrow) and numerous calcareous fossil fragments (thin white arrow) disseminated in the groundmass. Note: entire foraminiferal test (thick white arrow), (**b**) biogenic calcite debris: disarticulated rhodolith fragment (white arrow) and echinoid spine (green arrow), and crystal pyroclasts of quartz (yellow arrow) and biotite (blue arrow), (**c**) well preserved calcareous fossils: rhodoliths and foraminifers (white arrow). Note: presence of crystal pyroclasts: wedge-shaped quartz crystal fragment (yellow arrow) and biotite (blue arrow), (**d**) bryozoan fragment (thick white arrow). Note: crystal pyroclasts of quartz (yellow arrows): splinter-shaped crystal fragment (on the right) and a single grain with highly rounded outline (on the left).

The quartz occurs along with biotite being both disseminated in the groundmass (Figure 6a) and concentrated in laminae (Figures 6b,c and 7c,d). The disseminated, tiny quartz grains are less than 0.02 mm in size, whilst the larger ones, concentrated in laminae, reach up to 0.2 mm in diameter. Quartz grains are variable in shape; however, wedge- and splinter-shaped crystal fragments (Figures 6b,c, 7c,d and 8d) are the most common. Rounded and bipyramidal quartz crystals, rarely embayed, do occur, but less frequently (Figures 6c and 8d, respectively).

The biotite flakes, which lie approximately parallel to the bedding and laminae, defined by the larger biotite and quartz grains (fine sand in size), were only observed in the bottom and in the middle portion of the bed (sample B-1 and B-9; Figures 6 and 7). At its top (sample B-17), internal layering is hard to distinguish, the groundmass is less abundant and besides biotite and quartz, biogenic calcite detritus widely occurs (Figure 8).

The biogenic calcite detritus varies in size and in the state of its preservation. The grains are poorly sorted and unevenly disseminated in the smectite-composed groundmass. Besides disarticulated and comminuted calcareous organisms (Figure 8), whole foraminifer tests occur (Figure 8a,c). The most common biogenic calcite grains are red algae fragments up to 0.5 mm in size, which also vary in shape (Figure 8b), and localized accumulations of well-preserved rhodoliths up to 5 mm in diameter (Figure 8c). Besides the red algae,

calcareous foraminifers, which range from 0.05 to 0.2 mm in size, can easily be spotted in the groundmass (Figure 8a,d). Their chambers appear to be usually empty. Large bryozoan fragments (up to about 1 mm in size, Figure 8d), echinoid spines (about 0.2 mm in diameter, Figure 8b) and echinoderm ossicle fragments (up to a few millimeters in dimension) are rare.

*5.4. Field Emission Gun Scanning Electron Microscopy (FESEM)*

The FESEM studies revealed the texture of the groundmass in the rock studied, the degree of alteration of its original vitric constituents and the morphology of the resulting smectite, which was formed at the expense of the glass shards (Figure 9). In addition, the imaging allowed us to observe the shape, surface and alteration of non-clay minerals in order to identify igneous rock phenocrysts (Figure 10), to discover the internal texture of the biogenic detritus and its relationships to the altered pyroclasts (Figure 11) and to decipher the origin of iron oxide/oxyhydroxides, which occur along with the above-mentioned constituents in the clay bed (Figure 9e,f, Figures 10b and 11e,f).

The groundmass appears composed of granular fragments formed of smectite, the largest of which are less than about 100 μm in diameter and the most common are several micrometers in size. Various shapes of the granules were observed. There are spherical grains up to about 5 μm in diameter (Figure 9c,d) and elongated tubular aggregates less than 100 μm in length (Figure 9a,b,e,f). Irregularly shaped granules ranging from 5 to 100 μm in dimension are rare. The tubular aggregates are closed (Figure 9a,b,e,f) and open (Figure 9c) in character. They often form clusters up to several hundred micrometers in size. The texture of the groundmass seems to be inherited from a vitric ash. Unfortunately, smectite is the only constituent of all granular fragments, and no preserved glass shards were recognized. In the granules, tangential contacts between the smectite flakes are common (Figure 9f); less frequently, the flakes are arranged in a boxwork pattern with an orientation perpendicular to each other (Figure 9d).

The results of elemental analysis of the smectite obtained by using the FESEM/EDS device are shown in Figure 9d and are listed in Table 1. Based on these data, the structural formula of the smectite was calculated, which may suggest that the layer charge in this clay is mainly located in its octahedral sheets. This assumption is supported by the chemistry of the tetrahedral sheets in which the degree of Al for Si substitution (0.09) is low when compared to the amount of the charge compensating cations (i.e., Ca and Mg) in the interlayer space of the smectite (0.40) (Table 1).

Biotite often shows not only strong ductile deformations, which are visible at an optical microscopy scale, but also reveals numerous micro- to nanocracks oriented perpendicularly to the basal cleavage of this mica (Figure 10a). Due to the presence of the fractures, the biotite flakes are disarticulated, and their fragments reach even less than 2 μm in dimensions. Slightly distorted flakes (Figure 10b,d) and those which display no visible deformation (Figure 10c) are less common. The dimensions of the biotite grains measured perpendicular to their basal planes are usually about 20 μm, but a few thicker individuals were also spotted (Figure 10b). The biotite is Mg-Fe-bearing (Figure 10d), and iron prevails over magnesium in the structure of this mica (Table 2). In the vicinity of some biotite flakes, minute concentrations of iron oxides/oxyhydroxides were visible (Figure 10b).

The quartz grains vary in shape (Figure 10d–f) and range in size from a few to about 100 μm. There are crystal fragments, euhedral crystals and grains with highly rounded outlines. The crystal fragments display very sharp edges and conchoidal fracture surfaces. They are splinter- and wedge-shaped (Figure 10e). The euhedral crystals are tiny, double terminated beta-form quartz paramorphs having smooth crystal faces and rounded edges (Figure 10f). Grains with highly rounded outlines have a smooth surface like the euhedral crystals, but they are spherical or slightly ellipsoidal in shape (Figure 10d).

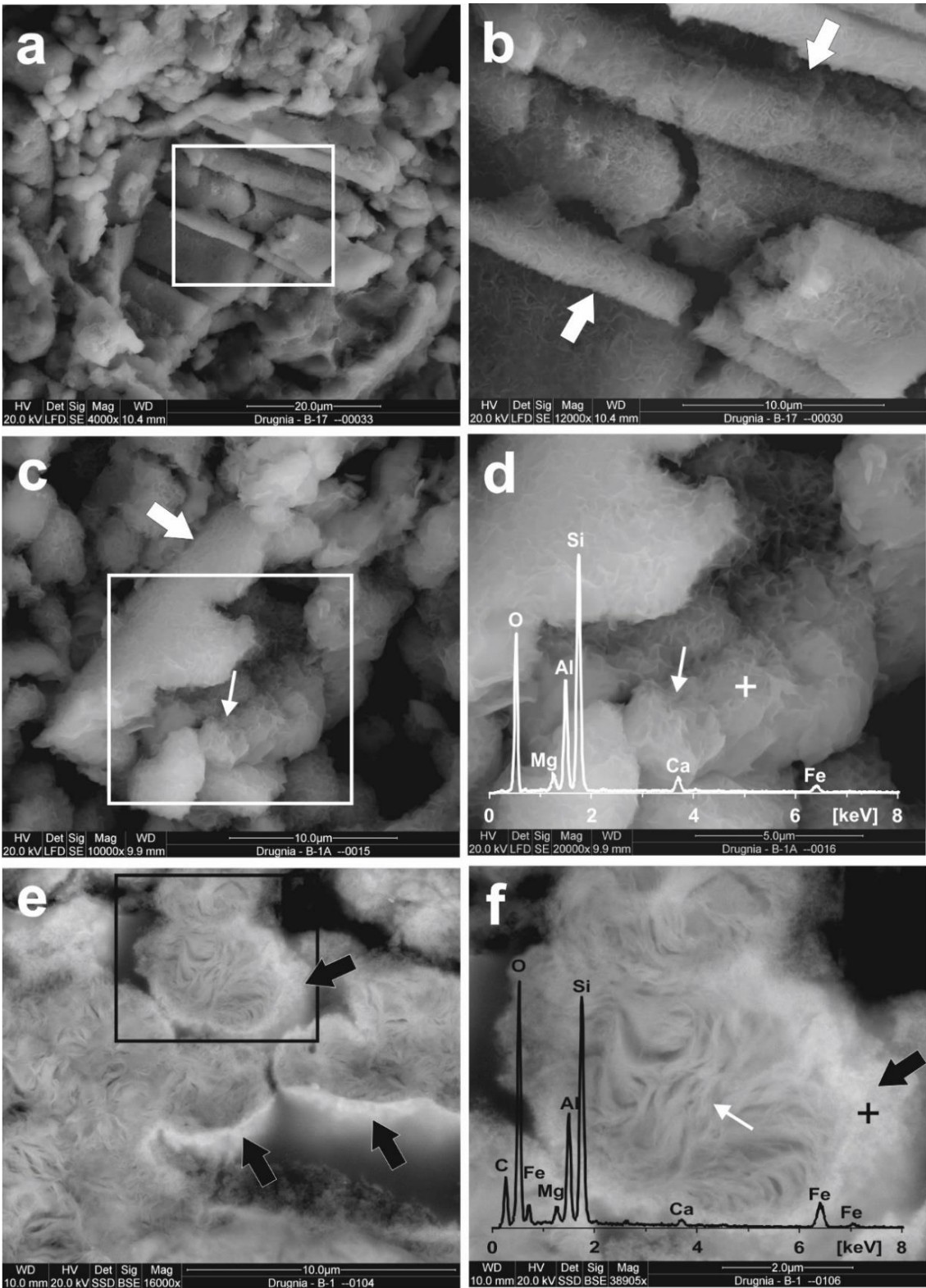

**Figure 9.** FESEM images of background composed of smectite in the Drugnia Rządowa bentonite ((**a**–**d**) FESEM/SE, rock chips and (**e**,**f**) FESEM/BSE, polished thin sections): (**a**) cuspate and curved-platy volcanic glass shards replaced by smectite. Note: clay material retained textures originally developed in glassy ash fragments (sample B-17); (**b**) enlargement of the area within the white box in (**a**) showing tubular aggregates of smectite (arrows); (**c**) smectite pseudomorph after curved-platy volcanic glass shards (thick arrow) and spherical aggregates of smectite (thin arrow) (sample B-1); (**d**) enlargement of the area within the white box in (**c**) showing boxwork texture of spherical aggregates of smectite (arrow). EDS data shows the chemical composition of

smectite measured in the area indicated by the white cross; (**e**) internal texture of tubular aggregates of smectite. Note: iron oxide/oxyhydroxide nanospherules forming crust on the outer surfaces of the aggregates (arrows) (sample B-1); (**f**) enlargement of the area within the black box in (**e**) showing tangential contact between smectite flakes in the interior of aggregates (white arrow). EDS data show the presence of iron in the crust of the smectite aggregates (black arrow) measured in the area indicated by the black cross.

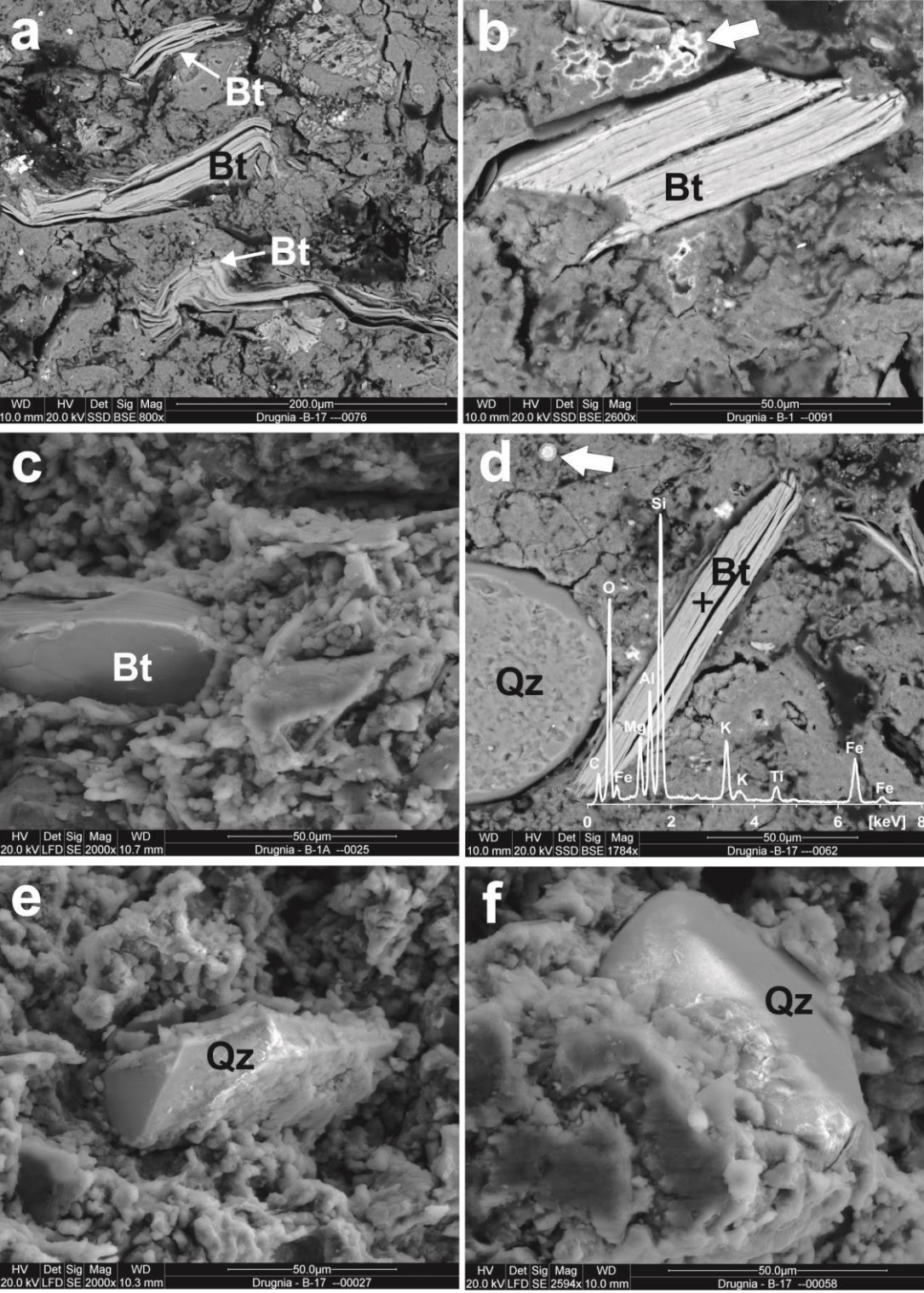

**Figure 10.** FESEM images showing a suite of crystal pyroclasts and the rate of their alterations in the Drugnia Rządowa bentonite ((**a,b,d**) FESEM/BSE, polished thin sections and (**c,e,f**) FESEM/SE, rock chips): (**a**) thin flakes of biotite (Bt) in smectite-composed groundmass. Note: alignment, ductile deformations and fracturing of the flakes (sample B-17); (**b**) thick slightly deformed flake of

biotite (Bt). Note: iron oxide/oxyhydroxide (arrow) accumulated on smectite aggregates in the vicinity of the biotite flake (sample B-1); (**c**) fresh, non-deformed flake of biotite (Bt) among smectite pseudomorphs after volcanic glass shards (sample B-1); (**d**) slightly deformed flakes of fresh biotite (Bt) and quartz crystal (Qz) showing a highly rounded outline in smectite-composed groundmass. EDS data show the chemical composition of biotite measured in the area indicated by the black cross. Note: spherical aggregates of iron oxide/oxyhydroxide (arrow) (sample B-17); (**e**) wedge-shaped quartz crystal fragment (Qz) showing a conchoidal fracture and sharp edges among smectite pseudomorphs after glassy pyroclasts (sample B-17); (**f**) bipyramidal volcanic quartz crystal (Qz) showing smooth edges embedded in smectite-replaced glassy pyroclasts. Note: very smooth surface of the crystal (sample B-17).

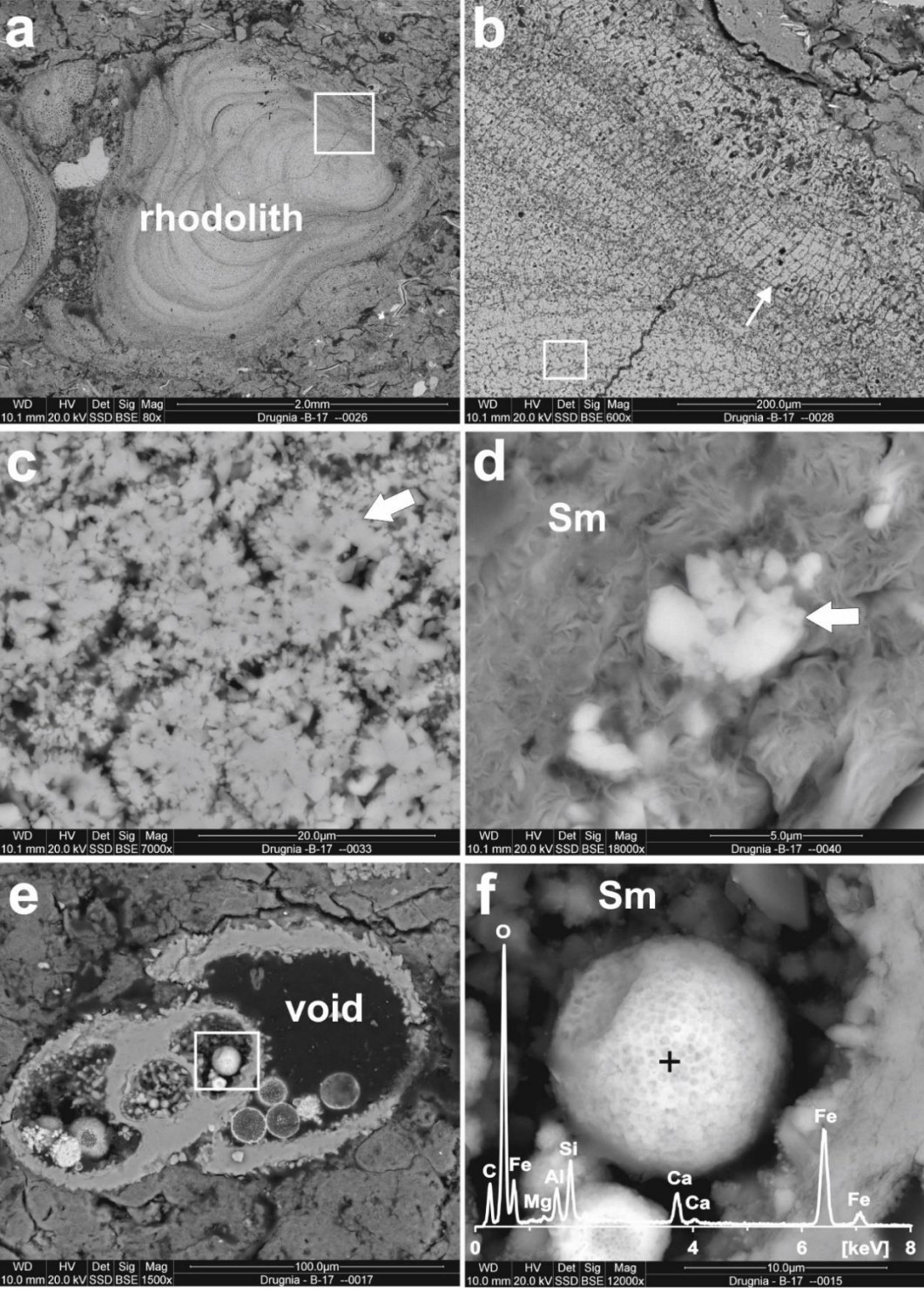

**Figure 11.** FESEM images showing biogenic calcite debris and its relationships to altered pyroclasts in the Drugnia Rządowa

bentonite (FESEM/BSE, polished thin section, sample B-17): (**a**) rhodolith; (**b**) enlargement of the area within the white box in (**a**), revealing palisade texture of coarse calcite crystals (arrow) in the outer part of the rhodolith; (**c**) enlargement of the area within the white box in (**b**) showing minute calcite crystals occurring in the inner part of the rhodolith arranged in rose-like aggregates (arrow); (**d**) rhodolith fragment: rose-like aggregate (arrow) arranged by minute calcite crystals embedded in smectite-composed groundmass (Sm); (**e**) foraminiferal test composed of calcite and partially filled with iron oxide/oxyhydroxide forming spherical aggregates variable in size and internal texture; (**f**) enlargement of the area within the white box in (**e**) showing a spherical aggregate of iron oxide/oxyhydroxide with particularly retained morphological attributes of a pyrite framboid (cellular-like texture). Note: chamber-lining spherical aggregates of smectite (Sm). EDS data show the chemical composition of the iron bearing spherical aggregate associated with clay measured in the area indicated by the black cross.

**Table 1.** FESEM/EDS analyses (wt %) and averaged structural formulae of smectite from the Drugnia Rządowa bentonite.

| Analyses | $SiO_2$ | $Al_2O_3$ | $Fe_2O_3$ | MgO | CaO | Total |
|---|---|---|---|---|---|---|
| | | | Sample B-1 | | | |
| Point 001 | 66.13 | 25.3 | 2.24 | 3.98 | 2.35 | 100.00 |
| Point 031 | 67.90 | 23.54 | 2.36 | 4.05 | 2.15 | 100.00 |
| Point 061 | 68.60 | 23.42 | 1.92 | 4.16 | 1.90 | 100.00 |
| | | | Sample B-9 | | | |
| Point 022 | 68.60 | 22.14 | 1.64 | 4.12 | 1.92 | 100.00 |
| Point 031 | 67.51 | 22.14 | 2.55 | 4.27 | 3.53 | 100.00 |
| Point 032 | 66.51 | 23.11 | 4.47 | 4.04 | 1.87 | 100.00 |
| Point 057 | 68.13 | 21.67 | 2.59 | 5.88 | 1.73 | 100.00 |
| | | | Sample B-17 | | | |
| Point 016 | 65.65 | 26.71 | 1.73 | 3.27 | 2.64 | 100.00 |
| Point 031 | 64.58 | 26.71 | 2.5 | 3.12 | 3.09 | 100.00 |
| Point 045 | 63.05 | 26.83 | 3.71 | 2.92 | 3.49 | 100.00 |
| Point 067 | 65.95 | 23.92 | 1.49 | 4.26 | 4.38 | 100.00 |
| Point 085 | 67.92 | 24.47 | 1.01 | 4.04 | 2.56 | 100.00 |
| Average | 66.71 | 24.29 | 2.35 | 4.01 | 2.63 | 100.00 |
| | | Averaged structural formulae based on 11 oxygen | | | | |

| Si | $Al^{IV}$ | $Al^{VI}$ | $Fe^{3+}$ | Mg | VI cations | Ca | Mg | Layer charge | Sum of the charges of the interlayer cations |
|---|---|---|---|---|---|---|---|---|---|
| 3.91 | 0.09 | 1.59 | 0.10 | 0.31 | 2.00 | 0.16 | 0.04 | 0.40 | 0.40 |

**Table 2.** Chemical analyses (FESEM/EDS) of biotite flakes from the bottom (sample B-1), middle portion (sample B-9) and the top (sample B-17) of the Drugnia Rządowa bentonite bed (wt %).

| Analyses | $SiO_2$ | $Al_2O_3$ | $Fe_2O_3$ | $TiO_2$ | MgO | $K_2O$ | Total |
|---|---|---|---|---|---|---|---|
| | | | Sample B-1 | | | | |
| Point 020 | 47.15 | 16.6 | 18.19 | 3.73 | 9.02 | 5.31 | 100.00 |
| Point 065 | 45.28 | 16.17 | 18.83 | 4.10 | 8.80 | 6.82 | 100.00 |
| Point 091 | 45.88 | 15.76 | 18.44 | 3.95 | 9.19 | 6.77 | 100.00 |
| Point 111 | 48.63 | 17.90 | 19.04 | 4.01 | 8.98 | 1.45 | 100.00 |
| Point 113 | 47.71 | 16.60 | 16.14 | 3.73 | 9.23 | 6.58 | 100.00 |
| Point 117 | 43.38 | 16.29 | 19.83 | 4.39 | 9.08 | 7.02 | 100.00 |
| | | | Sample B-9 | | | | |
| Point 004 | 46.43 | 16.11 | 17.34 | 3.87 | 9.26 | 6.99 | 100.00 |
| | | | Sample B-17 | | | | |
| Point 061 | 46.31 | 15.47 | 18.23 | 3.88 | 8.54 | 7.57 | 100.00 |
| Point 068 | 44.12 | 15.83 | 18.53 | 4.09 | 9.37 | 8.06 | 100.00 |

At the top portion of the clay bed studied, besides the above-mentioned constituents, a biogenic detritus, which is composed of calcite crystals variable in size and shape, was found. The red algae, bryozoans, and echinoid fragments, as well as entire foraminifers (Figure 11) can be distinguished. Red algae are composed of two types of calcite micro-crystals. The coarser of them, blocky in shape and of about a dozen micrometers in size,

are arranged parallel to the rhodolith outlines and occur in their outer parts (Figure 11b). Finer microcrystals of about 2 μm in size are visible inside of the rhodoliths and create rosette-like aggregates, which reach a dozen micrometers in diameter (Figure 11c). The fragments of disarticulated rhodoliths such as single rosette-like calcite aggregates and single crystals separated from them are common and can be spotted evenly distributed over the top portion of the clay bed. The fragments of bryozoans and echinoids, as well as foraminifers, are composed of calcite, which is massive-looking (Figure 11e,f). Only fragments of bryozoans and echinoids occur, while the whole tests of the foraminifers were well preserved (Figure 11e). The foraminifer chambers usually remained empty; only some of them are filled with smectite and with spherical aggregates of iron oxides/oxyhydroxides (Figure 11e,f).

The spherical aggregates composed of iron oxides/oxyhydroxides slightly vary in their shape and size. They are massive in appearance or show a cellular-like texture (Figure 11f) preserved to various degrees. Both aggregate types range in diameter from of one to a dozen micrometers. They usually occur accumulated inside of foraminifer chambers (Figure 11e,f). Less frequently, they were spotted as scattered in the smectite-composed groundmass (Figure 10d). In addition to the large spherical aggregates, localized minute nanospherules of iron oxides/oxyhydroxides were observed as concentrated on the edges of the smectite flakes, creating some clay granules that occur in the groundmass in the rock studied (Figure 9e,f). These nanospherules were also found as forming a pore-lining cement in the pore-space between some clay granules in the groundmass (Figure 10b).

## 6. Discussion

### 6.1. Preservation of Volcanic Markers in the Clay Interbedded with Lithothamnium Limestone in the Drugnia Rządowa

Although the main evidence for volcanic origin is afforded by the presence of volcanic glass shards, the Drugnia Rządowa clay bed contains vitric fragments completely altered to clay that retained characteristic textures inherited from volcanic glass (Figure 9). A modified glass-relic texture was often observed in bentonites containing nothing but clay material. The microfabric of altered ash layers, similar to those developed in the clays studied, was described by Tribble and Wilkens [5] and Li et al. [46]. It is worth highlighting that the smectite granules which constitute the groundmass of the rock studied retained the glass-relic textures with surprising perfection. The tubular aggregates especially reveal the shape and the size attributed to the plate, curved plate and cuspate volcanic glass shards, which were slightly distorted because of the alteration to smectite (Figure 9). It is thought that the platelike glass shards result from the almost complete fragmentation of the glass by explosive action and are in large part the remnants of the walls that surrounded the flat lenticular and more or less spherical bubbles (see [4]).

The volcanic origin of the clays can be also confirmed through the recognition of a simple suite of non-clay minerals, which show crystal habits characteristic of pyroclastic rocks, i.e., fragmented or euhedral phenocrysts of volcanic origin. The term "phenocrysts" is used in this paper after Huff et al. [47] in order to describe: "juvenile crystal pyroclasts which are products of parental magma crystallization and have been emplaced as discrete fallout particles". The clays studied comprise fragmented and euhedral phenocrysts of biotite and quartz (Figures 6, 7c,d and 10) scattered in the groundmass composed of smectite pseudomorphs after volcanic glass shards, as described above. The volcanic origin of the biotite is confirmed by the presence of strong, both ductile and brittle deformations of its flakes (folding and fracturing) (Figures 6d and 10a). Such distortions are thought to result from an explosive eruption with the rapid decline in pressure and thermal shock of the phenocrysts. The crystal habit of the quartz, i.e., euhedral, doubly terminated crystals (Figure 10f), rarely embayed (Figure 6c), crystals showing highly rounded outlines (Figure 10d) and splinter- and wedge-shaped crystal fragments (Figures 6b,c, 7c,d, 8b–d and 10e) correspond to those referred to volcanic (pyroclastic) characteristics described by, e.g., Bohor and Triplehorn [7] and Fisher and Schmincke [4].

The nature of the smectite, which occurs in the bed studied, corresponds to the nature of a smectite, which is usually formed at the expense of volcanic glass fragments. Smectite resulting from the alteration of volcanic glass in pyroclastic rocks was described by, e.g., Khoury and Eberl [48], Christidis [49] and White et al. [50], and that obtained by laboratory experiments was reported by, e.g., Fiore et al. [51]. The smectite that replaces glass both in a natural environment and in laboratory conditions shows characteristic authigenic clay mineral growths. The imaging of the smectite from the Drugnia Rządowa clay bed (Figure 9b–f) revealed authigenic smectite attributes (see [52]). In addition to the authigenic appearance, the crystal chemistry of the smectite studied (Table 1) also seems to be related to the process of clay formation due to alteration of volcanic glass fragments, and the resulting products were protected against later diagenetic changes. The crystal chemistry of smectites originated from transformation of volcanic glass were discussed in the papers by, e.g., Deconinck and Chamley [53] and Drits and Zviagina [54]. As far as the clay studied is concerned, because the XRPD examination showed that the $d_{003}$ spacing of the smectite mineral is 5.60 Å (Figure 5), this mineral can be considered as being composed of only swelling layers. Consequently, based on the nature of the smectite mineral, the diagenetic changes recognized in the clay bed studied seem to be not very advanced. The texture originally developed in glassy ash was not obliterated by the mechanical compaction, and the burial temperature was lower than that required to initiate the smectite to illite transition. This finding is supported by the presence of perfectly preserved rhodoliths (Figures 8c and 11a–c) and only slightly altered internal textures of remnant fossils (Figures 8b,d and 11e,f) observed at the top portion of the clay bed studied, as well as being documented by the stratigraphic evidence (e.g., [30]). Thus, the smectite studied was not remarkably modified during burial diagenesis if it could be accepted that its crystal chemistry developed during its formation due to the alteration of volcanic glass. Because only 22% of charge originated from the tetrahedral sheets (Table 1), the smectite from the Drugnia Rządowa clay bed is a montmorillonite. It should be mentioned that the procedure of the layer charge calculation presented in this paper considers the limitation of the FESEM/EDS method and possible contribution of pyroclastic crystals such as biotite and quartz, and iron oxides/oxyhydroxides in the overall chemical composition. A montmorillonite, often highly crystallized, is generally accepted as the most common clay originating from the alteration of volcanic glass in a marine environment (e.g., [3,55]). The intense and relatively narrow (FWHM = 0.58° 2θ) basal peak of the montmorillonite from the clay bed studied suggests that this clay is highly crystallized. The described-above attributes of the clay bed studied clearly indicate that the clay originated from in situ alteration of pyroclastic material. Therefore, the Drugnia Rządowa clay bed characteristics meet the criteria of the commonly accepted definition of bentonite published in Ross and Shannon [56].

The iron oxides/oxyhydroxides are secondary components of the bentonite studied and were sourced from the alteration of the biotite flakes. The accumulation of these components in the vicinity of the biotite flakes confirm the redistribution of iron liberated from the lattice of this phenocryst (Figure 10b). Only in the top portion of the bentonite bed, spherical aggregates composed of iron oxides/oxyhydroxides suggest that they were sourced from the alteration of pyrite framboids. Because these iron-bearing microspheres seem to be accumulated in the manner attributed to pyrite framboids, and are often associated with fossils (Figure 11e,f), they were submitted to the pyroclastic material along with biogenic calcite debris.

### 6.2. Peculiarity of the Bentonite Bed

The microscopic texture of the bentonite studied, which consists of crystal pyroclasts of biotite and quartz scattered through a groundmass of secondary montmorillonite formed at the expense of platy and cuspate volcanic glass shards (Figure 10c,e,f), clearly demonstrate the origin of the bed related to a fallout of discrete particles. It should be remembered here that the occurrence of bentonite beds hosted by the lithothamnium limestone of the

Pińczów Fm. was never reported in the scientific papers on the geology of the northern part of the Carpathian Foredeep. Thin interbeds of bentonites were spotted in the deposits underlying and overlying the Pińczów Fm. The former formation is called the Trzydnik Fm. and occur only in the Korytnica Bay, whilst the latter are known as the Chmielnik Fm./Machów Fm. and crop out in the forebay zone (Figure 1) (e.g., [20,24,39,40]). Until now, in the area of the Pierzchnica Bay, where the outcrop studied is located (Figure 1), bentonites have not been discovered.

A specific, noticeable feature of the bentonite studied is the very fine grain size of its primary volcanogenic constituents. The average dimension of the montmorillonite pseudomorphs replacing volcanic glass shards is approximately 20 μm, and its maximum size reaches 100 μm (Figure 9). The size of crystal pyroclasts is similar to the estimated dimensions of the glass fragments (Figures 6a,c, 7c,d and 10d–f), except for some biotite flakes, which are larger and even reach 400 μm in diameter (Figures 6b,d and 10a). Micrometer-sized pyroclasts, which were the primary constituents of the Drugnia Rządowa bentonite, can be considered as indicators of a distal location of its source volcanoes.

The distinctive attributes of the bentonite bed studied are also its noticeable thickness (0.6 m—Figure 3) and a very simple mineralogy (Figure 4). The bed contains almost nothing but montmorillonite. The montmorillonite clay accounts for approximately 97%, almost 99% and approximately 90% of the constituents in the bottom, middle and top portions of the bed, respectively. Because the biogenic calcite detritus (i.e., local background sediments) has been identified only at the top portions of the bed (Figure 4, sample B-17; Figures 8 and 11), it can be considered that the bed was primarily composed exclusively of glassy and crystal pyroclasts. The submission of biogenic calcite detritus probably followed rather than accompanied the deposition of volcanic particles. The data presented in this paper indicate that the mineralogy varies only slightly over the thickness of the bentonite layer; therefore, it can be suggested that the Drugnia Rządowa bed probably resulted from a single ash fall. The montmorillonite clay-dominated composition of the bentonite documents that mostly volcanic glass fragments were deposited. Consequently, the precursor pyroclastic rock for the Drugnia Rządowa bentonite can be called a vitric ash. Because the background-originated material was spotted only at the top of the bentonite bed, the presence of laminae defined by the coarser crystal pyroclasts in its bottom and middle part probably resulted from the mode of pyroclasts deposition (i.e., closely spaced "pulses" of pyroclasts within a single event). Similar laminae were demonstrated by, e.g., [4,46]. The suite of crystal pyroclasts (biotite and quartz) indicates that the precursor pyroclasts for the bentonite studied was produced by a high-silica magma. In summary, the Drugnia Rządowa bentonite bed, which is hosted by the lithothamnium limestones from the northern margin of the Carpathian Foredeep, can be considered as a record of a large-scale explosive eruption of silicic magma chambers. This rock represents a thick bentonite bed resulting from the alteration of distal fall pyroclasts, which rarely occur well preserved in the geological record. It should be stressed that thick bentonite is especially useful for interpreting the geodynamic evolution of our planet and is also regarded as a valuable raw material. Therefore, its components, especially clay minerals, draw the attention of researchers. Some examples of widely known thick bentonite beds are the Ordovician Deicke and Millbrig K-bentonites in North America and the Kinnekulle K-bentonite in northwestern Europe [47], the Permian Melo bentonite in Uruguay [55] and the Cretaceous Charente bentonite in France [57].

### 6.3. Possible Location of the Source Volcanoes

When accepted that the average grain size of the primary pyroclastic debris, which were deposited in the sedimentary basin of the lithothamnium limestones from the Drugnia Rządowa, refers to medium dust (5–6 Φ, 32–16 μm), their source volcanoes could be located at a maximum distance of approximately 300–400 km from this basin. This estimate is based on the diagram presented in Fischer and Schmincke [4] (p. 155). Owing to this estimation of distance, it is reasonable to assume that the volcanoes from the inner part

of the Carpathian arc sourced the precursor pyroclastic deposits for the bentonite bed studied. The source of the pyroclastic material deposited in the Carpathian Foredeep basin located in the inner arc of the Carpathians was already suggested by Tokarski [58]. The suggestion of this author was supported by volcanic glass chemical analyses indicating acid volcanism, and grain size analyses of pyroclasts implying the transport of material from the south. The volcanoes accommodated in Slovakia were situated at the above-mentioned distance from the northern margin of the Carpathian Foredeep, in the south. Therefore, the Central Slovakia Volcanic Field (CSVF), which is the largest volcanic area in the Western Carpathians, could be considered as the source volcanoes for the fallout ash deposited during the sedimentation of the lithothamnium limestones at the northern margin of the Carpathian Foredeep. Violent explosive eruptions of volcanoes in this area might have been able to produce thick distal ash deposits. The volcanoes of the CSVF were active from the early Badenian to late Sarmatian [59]. Although the CSVF seems to be the most probable source of the pyroclastic material because of its size and location, the volcanic fields located in Hungary (Bükk Foreland, Tokaj and Zempléni Mountains, e.g., [13,14,60]) cannot be excluded.

As mentioned above, the presence of biotite and quartz pyroclasts demonstrate evidence that the glass from which the bentonite studied derived was excessively rich in silica. The chemical composition of the biotite (Fe prevailing over Mg—Table 2) also confirms that the phenocrysts originated from silicic magma (e.g., [61]). In the CSVF area, the products of acidic volcanism are common. The bentonites, which were formed at the expense of acidic ignimbrites, were reported from the southern part of the CSVF (e.g., [22,62–67]).

*6.4. Some Remarks on the Use of the Bentonite as Raw Materials*

Although pyroclastic rocks with different degrees of alteration are common in the Miocene succession of the Carpathian Foredeep (see [16,17]), bentonites were formed mostly at the northern margin of the basin, probably due to the contrasting tectonic and sedimentary history of its northern and southern peripheral parts. Bentonite beds, which are thick enough to be used as a raw material, are present in the Carpathian Foredeep, but rare. Besides the bentonite studied, another example is the bentonite deposits from Opoka Duża (Figure 1). The latter was used in order to produce high-quality bleaching earth known under the trade name "Supersileton" [23]. Because the geological data on the Opoka Duża bentonite bed are scarce, it is difficult to determine its relationship to the bentonite studied. However, based on Czarnocki's [23] paper, it is reasonable to assume that both the Opoka Duża bed and the Drugnia Rządowa bed are hosted by coeval and equivalent lithologies, which refer to them as members of the Pińczów Fm. Consequently, geological exploration for economic bentonite at the northern margin of the Carpathian Foredeep focused on the Pińczów Fm. are worth undertaking. Probably a coeval pyroclastic layer (Wiatowice tuff described by Bukowski et al. [68]) deposited at the southern margin of the Carpathian Foredeep, in open marine conditions, unfortunately seems to be only slightly altered to clay. Taking into account the history of production and industrial application of the Opoka Duża bed, as well as the results of these studies on mineralogy of the Drugnia Rządowa bed, the bentonites hosted by the Pińczów Fm. can be regarded as a valuable raw material, especially owing to its prominent montmorillonite content. Bentonites are a highly demanded industrial commodity [9,11]. The shortage of high-quality bentonite deposits in Poland, and the need to protect our planet and its resources, are the main reasons why the proper use of valuable raw materials becomes especially important. Therefore, the bentonite studied, which consists almost exclusively of montmorillonite, can be regarded as a valuable industrial commodity and used, e.g., in nanotechnology. Because the bentonite bed is hosted by the lithothamnium limestones and exposed in the limestone open pit, it can be excavated as an associated deposit. Consequently, the excavation cost and the impact of the bentonite mining on the environment can be reduced.

### 7. Concluding Remarks

The petrographic features (e.g., texture and mineralogy) clearly evidence that the clay bed hosted by the lithothamnium limestones of the Pińczów Fm. (Badenian) is a bentonite in origin. This bentonite, called the Drugnia Rządowa bentonite bed, was formed due to the accumulation and alteration of distal airborne glassy dust and subordinate single crystals, and has never been deeply buried. The bentonite largely consists of highly crystallized montmorillonite. The crystal pyroclast suite (biotite and quartz) indicates this bentonite affinity for acidic volcanism.

The noticeable thickness (0.6 m) and prominent content of highly crystallized montmorillonite, as well as the fine grain size of scarce crystal pyroclasts, are the main attributes of the Drugnia Rządowa bentonite bed. These attributes make the bed a high-quality commodity, which can be used in technologies that require special raw materials.

The occurrence of the thick fallout ash bed hosted by the lithothamnium limestones can be regarded as a record of one of the most extensive explosive volcanic eruptions in the late Miocene. This eruption, which violently interrupted littoral carbonate sedimentation at the northern margin of the Carpathian Foredeep, is reported for the first time in this paper.

The results of these studies highlight the level of volcanic activity related to the closure of the Outer Carpathian basin, which has been recorded in the Miocene succession of the Carpathian Foredeep. Numerous thin interbeds of pyroclastic rocks, which have been reported in the literature, evidence the high frequency of volcanic activity in this region. This paper for the first time reports the occurrence of a thick bentonite bed in the Polish part of the Carpathian Foredeep, which documents a single, very violent volcanic eruption. In addition to that, this paper also reveals for the first time the record of the volcanic activity retained in the lithothamnium limestones. Therefore, these data could be important for the reconstruction of the geodynamic evolution of the Carpathian Foredeep margin. However, the absolute age of the Drugnia Rządowa bentonite bed and its regional correlation, as well as the location of possible source volcanoes, require more detailed studies, which will be published in the near future.

**Author Contributions:** Conceptualization, K.G., T.S. and P.W.; field work and sampling, methodology, K.G., T.S. and P.W.; investigation, K.G., T.S., A.G. and M.N.; writing—original draft preparation, K.G. and T.S.; writing—review and editing, K.G., T.S., P.W. and A.G.; visualization, K.G. and T.S. All authors have read and agreed to the published version of the manuscript.

**Funding:** This work was supported in part by a subsidy 16.16.140.315 from the AGH University of Science and Technology, Faculty of Geology, Geophysics and Environmental Protection, Krakow, Poland. The paper also includes, in part, the results obtained owing to the financial support of the Faculty of Geology, Geophysics and Environmental Protection used for laboratory work performed during 2020–2021 in order to prepare the Engineering Diploma Thesis by Małgorzata Niemiec under supervision of Katarzyna Górniak.

**Data Availability Statement:** All data are included in the paper.

**Acknowledgments:** We would like to thank the Kopaliny Mineralne sp. z o.o. company for allowing us on site to conduct a field study and to sample bentonite exposed within the open pit. The staff of the Drugnia Rządowa open pit is gratefully thanked for their assistance with the sampling. We also very much appreciate the help of Justin Quilling for his help with the English grammar and writing style. The authors thank the reviewers for their remarks and comments.

**Conflicts of Interest:** The authors declare no conflict of interest.

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
