# Peer review of "Recently Discovered Thick Bentonite Bed Hosted by the Lithothamnium Limestones (Badenian) in the Polish Part of the Carpathian Foredeep: The Evidence for Volcanic Origin"

_minerals, doi:10.3390/min11121417_

Round 1
Reviewer 1 Report
This paper is very wonderful. The same method can be used in the study of other geological ages. It just needs some supplement. The crystal pyroclast suite (biotite and quartz) should be sorted out and observed by scanning electron microscope whether it is the source of pyroclastic. In addition, it can be sorted by heavy minerals. Can zircon be selected and dated?
Reviewer 2 Report
The manuscript is mostly well written, there are some mistypes and some strange language use, therefore, a thorough language check is needed.
The text is a bit lengthy, i.e. there are repeated parts (i.e. in the Methods section the authors refer in advance to the results and discussion parts) which could be left out, and the text could be shortened.
The manuscript contains detailed information on the studied samples which are well documented and clearly interpreted. My only concerns are related to the geological background knowledge about these pyroclastic layers and to the regional interpretations of them. I lack some publications which deal with similar bentonite deposits from the surroundings of the Carpathians (Carpathian-Pannonian region) such as Rocholl et al, 2018 or see the summary of Lukács et al., 2018 and references therein. Miocene bentonite deposits are known from several localities, which were interpreted so far as being related to the Miocene silicic volcanism in the Carpathian-Pannonian region. We already know a lot about the timing of these bentonite deposits and their correlations with proximal deposits although there are still a lot of questions. This could be emphasised in the Introduction to give a greater importance of the findings.
In line 65-67, for getting knowledge on the volcanism of the inner part of the Carpathian arc the reader is suggested to get the paper of Lexa et al. 2010, however, there are a lot more which dealt with the volcanism of the area: e.g. Szabó et al., 1992; Harangi et al., 2007; Harangi and Lenkey, 2007; Seghedi et al. 2004; 2005, Pécskay et al., 2006; Lukács et al., 2018, etc... It is also important to note that such large volume distal bentonites are usually linked to high-silica eruptions (which ash material can travel higher distances), therefore, I suggest to highlight such volcanic eruptions/events in the area.
In line 67-69, the authors cite a review article that summarizes the pyroclastic interbeds in the Polish part of the Carpathian Foredeep. The main problem with this is that it is not written in English, therefore, the reader (if not Polish) got zero information about these. So, here I suggest to give a figure with the pyroclastic horizons and/or a text paragraph dealing with the results of the review.
The authors are citing some references of bentonite and pyroclastic deposits from the Polish Foredeep, which are mostly in Polish, so again a foreigner reader is not necessarily able to read these works. As far as I know, there are some papers in English which could be referred and should be placed in the background story (i.e. Bukowski et al., 2010 and 2018), as well.
Section 6.3.
You may also consider the possibility of sourcing the bentonites from other areas than CSVF having Badenian-Sarmatian age, i.e. there are silicic volcanic fields with extensive pyroclastic deposits in the Bükk Foreland (Szakács et al., 1998; Lukács et al., 2015; 2018; 2021) and also in the Tokaj Mts. (e.g. Pécskay et al., 2006; Szepesi et al., 2019).
In line 618-619, the authors declare that Tokarski (1953) suggested the Central Slovakian Volcanic Field (CSFV) as a source for the bentonites. Would be good to tell on what ground?
A comparison with the Slovakian bentonites similar to the one given in Górniak et al. 2017, based on mineralogy and chemistry could be added to the manuscript.
I am looking forward to having age results of the bentonite, which is planned to come in the near future.
Reviewer 3 Report
Dear Authors,
I have read with interest your paper “Recently discovered thick bentonite bed hosted by the lithothamnium limestones (Badenian) in the Polish part of the Carpathian Foredeep: the evidence for volcanic origin”. I have found your work interesting, especially for detecting for the first time in the area a bentonitic deposit that could be exploited. However, the paper has some weakness points that need to be amended before publication.
I list here the main points. Along the text I have annotated this and minor suggestions.
- The list of reference is not very up to date: there are only a very limited number of papers of the last 10 years, and most of the cited papers are well more of 20 years old.
- The clay layer is not well described. Also fig. 3, in which it is visible, could be bigger and with a more detailed caption.
- I am not an expert of XRPD analyses on clay minerals. However, as far as I know, thermic treatments at different temperatures and Mg and K treatments are usually done, together with the glycol treatment, in order to identify the clay minerals without ambiguities. Is there a reason for not doing it?
- The strongest weakness point is represented by the microanalyses: first of all, they are obtained by EDS, which is not the best method for hydrous minerals such as clay minerals or biotite, because it closes the sum of oxides to 100%. This is a problem because these minerals contain a high, yet undefined, amount of water. The lack of this important element and the closure to 100 affect the concentration of other oxides and, therefore, make the recalculation speculative. The second problem is that analyzed points are very few and all obtained on the same sample. The best would be to do a higher number of analyses, possibly by WDS. If this is not possible I suggest to delete the composition of biotite, which does not offer any additional information, and, when discussing the origin of smectites, use caution in the identification of minerals based on the recalculation of the formula. All the discussion should be based mainly on the more robust XRPD data.
- In the discussion (subsection 6.2, lines 593-595) the clay layer is interpreted as the result of alteration of a primary fall deposit, and the occurrence of laminae with different grain size as expression of pulses in the eruption. However, the thickness of the deposit is hardly compatible with that of a primary fall of a very distal eruption (as suggested by the small size of pyroclasts), while it is well explained by the reworking and accumulation of the material. The absence of non-volcanic fragments requires that this reworking is syn-eruptive. The variations in grain size and the concentration of minerals in preferential layers can be an effect of reworking as well. I therefore suggest to mention this alternative hypothesis.
- Sub section 6.3. The calculation of the distance from the vent based only on the grain size is not possible if other parameters (density of the particles, wind direction, height of the eruptive column etc.) are known. The calculated distance of 300-400 km can be considered as a maximum distance and the volcanoes falling within this range must be taken into account.
I hope you find these comments useful and that they help to increase the quality of your paper.

Round 2
Reviewer 3 Report
Dears Authors, I have read the revised version of your manuscript and your point-by-point reply to my comments. I appreciate the efforts done to improve the paper, in particular in the upgrading of the reference list and the add of new EDS analyses.
I still keep my doubts on the goodness of the SEM analysis, and, as suggested in the previous comments, I would reduce the importance to these data given in the discussion (in particular to the biotites, which are not particularly useful for your conclusions). I also suggest to revise the English form, beacause some typos or minor errors still occur (e.g.: the title of subsection 6.4 should be "Some..." instead of "Same..."). After these minor changes the paper can be accepted.
Author Response
Answers to the remarks of Reviewer 3 (Review 2)
Thank you very much for your comments and suggestions. All your suggestions were taken into consideration. The answers to your remarks are formulated below and the PDF with track changes is attached as separate file.
Answers to the Comments and Suggestions for Authors
Remark: I still keep my doubts on the goodness of the SEM analysis, and, as suggested in the previous comments, I would reduce the importance to these data given in the discussion (in particular to the biotites, which are not particularly useful for your conclusions).
Answer: We reduced the importance of the data on biotite given in the Discussion chapter as follows: “As mentioned above, the presence of biotite and quartz pyroclasts demonstrate evidence that the glass from which the bentonite studied derived was excessively rich in silica. The chemical composition of the biotite (Fe prevailing over Mg – Table 2) is also confirms evidence that the phenocrysts originated from silicic magma (e.g., [60]). In the CSVF area the products of acidic volcanism are common. The bentonites, which were formed at the expense of acidic ignimbrites, were reported from the southern part of the CSVF (e.g., [22, 61-66].”
Remark: I also suggest to revise the English form, beacause some typos or minor errors still occur (e.g.: the title of subsection 6.4 should be "Some..." instead of "Same..."). After these minor changes the paper can be accepted.
Answer: The mistakes in typing have been removed and minor errors have been corrected.
